# Natural Course of *IQSEC2*-Related Encephalopathy: An Italian National Structured Survey

**DOI:** 10.3390/children10091442

**Published:** 2023-08-24

**Authors:** Silvia Leoncini, Lidia Boasiako, Diego Lopergolo, Maria Altamura, Caterina Fazzi, Roberto Canitano, Salvatore Grosso, Ilaria Meloni, Margherita Baldassarri, Susanna Croci, Alessandra Renieri, Mario Mastrangelo, Claudio De Felice

**Affiliations:** 1Neonatal Intensive Care Unit, Department of Women’s and Children’s Health, University Hospital Azienda Ospedaliera Universitaria Senese, 53100 Siena, Italy; s.leoncini74@gmail.com (S.L.); lidiaboasiako@gmail.com (L.B.); maria.altamura1997@gmail.com (M.A.); caterina.fazzi@student.unisi.it (C.F.); 2Rett Syndrome Trial Center, University Hospital Azienda Ospedaliera Universitaria Senese, 53100 Siena, Italy; 3Department of Medicine, Surgery and Neurosciences, University of Siena, 53100 Siena, Italy; diego.lopergolo@unifi.it; 4UOC Neurologia e Malattie Neurometaboliche, Azienda Ospedaliero Universitaria Senese, Policlinico Le Scotte, 53100 Siena, Italy; 5IRCCS Stella Maris Foundation, Molecular Medicine for Neurodegenerative and Neuromuscular Diseases Unit, 56018 Pisa, Italy; 6Child Neuropsychiatry Unit, Department of Mental Health, University Hospital Azienda Ospedaliera Universitaria Senese, 53100 Siena, Italy; r.canitano@gmail.com; 7Department of Molecular and Developmental Medicine, University of Siena, 53100 Siena, Italy; salvatore.grosso@unisi.it; 8Pediatric Unit, Department of Women’s and Children’s Health, University Hospital Azienda Ospedaliera Universitaria Senese, 53100 Siena, Italy; 9Medical Genetics, University of Siena, 53100 Siena, Italy; ilaria.meloni@unisi.it (I.M.); margherita.baldassarri@dbm.unisi.it (M.B.); susanna.croci@dbm.unisi.it (S.C.); alessandra.renieri@unisi.it (A.R.); 10Med Biotech Hub and Competence Center, Department of Medical Biotechnologies, University of Siena, 53100 Siena, Italy; 11Genetica Medica, Azienda Ospedaliera Universitaria Senese, 53100 Siena, Italy; 12Maternal Infantile and Urological Sciences Department, Sapienza University of Rome, 00185 Rome, Italy; mario.mastrangelo@uniroma1.it; 13Child Neurology and Psychiatry Unit, Department of Neurosciences and Mental Health, Azienda Ospedaliero-Universitaria Policlinico Umberto I, 00161 Rome, Italy

**Keywords:** *IQSEC2* gene, Rett syndrome, X-linked intellectual disability, developmental encephalopathies, epilepsy, gut–brain axis, sleep disorders, co-sleeping

## Abstract

Pathogenic loss-of-function variants in the IQ motif and SEC7 domain containing protein 2 (*IQSEC2*) gene cause intellectual disability with Rett syndrome (RTT)-like features. The aim of this study was to obtain systematic information on the natural history and extra-central nervous system (CNS) manifestations for the Italian *IQSEC2* population (>90%) by using structured family interviews and semi-quantitative questionnaires. *IQSEC2* encephalopathy prevalence estimate was 7.0 to 7.9 × 10^−7^. Criteria for typical RTT were met in 42.1% of the cases, although psychomotor regression was occasionally evidenced. Genetic diagnosis was occasionally achieved in infancy despite a clinical onset before the first 24 months of life. High severity in both the CNS and extra-CNS manifestations for the *IQSEC2* patients was documented and related to a consistently adverse quality of life. Neurodevelopmental delay was diagnosed before the onset of epilepsy by 1.8 to 2.4 years. An earlier age at menarche in *IQSEC2* female patients was reported. Sleep disturbance was highly prevalent (60 to 77.8%), with mandatory co-sleeping behavior (50% of the female patients) being related to de novo variant origin, younger age, taller height with underweight, better social interaction, and lower life quality impact for the family and friends area. In conclusion, the *IQSEC2* encephalopathy is a rare and likely underdiagnosed developmental encephalopathy leading to an adverse life quality impact.

## 1. Introduction

Pathogenic variants on chromosome X represent an important cause of neurodevelopmental disorders (NDDs) [1]. Clinical features of X-linked NDDs are influenced by gender [2,3]. In female patients, clinical phenotype is more variable due to X chromosome inactivation (XCI) status [2,3] and the rate of genes escaping XCI (15–20%) [4], such as IQ motif and SEC7 domain containing protein 2 (*IQSEC2*) [5]. Variants in the *IQSEC2* gene are associated with a rare form of intellectual disability (ID) and epilepsy (OMIM # 300522) in males and females [5]. IQSEC2, a guanine nucleotide exchange factor able to activate small GTPases of the ARF family [5], acts on shaping dendritic spine morphology [6] and controls excitatory synaptic transmission by regulating glutamate (N-methyl-D-aspartate, NMDA, and α-amino-3-hydroxy-5-methyl-4-isoxazolepropionic acid, AMPA) receptor-mediated responses at the excitatory synapses [7,8]. To date, the prevalence of *IQSEC2*-related encephalopathy is unknown. Although about 100 cases with pathogenic variants of *IQSEC2* were described in the medical literature [9,10,11,12,13,14], the clinical phenotype has not yet been fully delineated. Pathogenic variants in *IQSEC2* account for approximately 2% of patients with ID and epilepsy referred for exome sequencing [15].

*IQSEC2* variants can affect both genders with a clinical phenotype generally more variable in affected females than males who usually present more severe symptoms [12,13,16].

A wide phenotypic spectrum for *IQSEC2*-related encephalopathy is reported. The clinical phenotype can include ID, hypotonia, moderate-to-severe delayed psychomotor development, poor speech, seizures, and stereotypic movements [12,14,17,18,19]. ID and developmental delay are reported to be more severe in males than in females [12,13]. Additional features may include autism spectrum disorder-like (ASD-like) [5,13,20,21] and/or psychiatric features (attention deficit hyperactivity disorder (ADHD)-like behaviour, aggressiveness, irritability, self-injurious behavior) [1,10,11,20,21]. Experiments using *IQSEC2* mouse models (i.e., A350V transgenic) and three knockout (KO) mouse models recapitulated the presence of autistic-like behavior and epileptic seizures in the affected animals [22]. In particular, gastrointestinal symptoms have been also reported in *IQSEC2*-related encephalopathy [10,11,14].

Rett syndrome (RTT) is widely recognized as a paradigmatic developmental encephalopathy [23], with several genetic syndromes reported to exhibit RTT-like phenotypes [24,25]. RTT (OMIM #312750) is a rare genetic disorder leading to severe and progressive intellectual disability affecting almost exclusively females (incidence 1:10,000), and is mainly linked to mutations in the gene encoding methyl-CpG-binding protein 2 (MeCP2) [26,27]. RTT alone accounts for 10% of cases of profound ID of genetic origin in females [28]. Following a period of seemingly early normal development (6–18 months of age), early signs are commonly represented by regression of early developmental milestones, onset of hand stereotypies and deterioration of motor skills, eye contact, speech, and motor control [29,30]. The main neurological signs include autonomic dysregulation, epilepsy, swallowing dysfunction, sleep dysfunction, abnormal movements, and behavior disturbances [29,30]. Sleep dysfunction (about 90% of individuals) is mainly characterized by difficulty falling asleep and/or staying asleep, with important impacts on overall family quality of life [31]. Seizures and anxiety are very common features in RTT [25,31,32] along with sleep problems and breathing abnormalities [33,34,35,36,37,38]. Wide variability in clinical phenotypes of RTT patients is known and often related to specific *MECP2* variants [39,40]. Initially considered a mere nervous system pathology, over the years, RTT has gained recognition as a multisystemic disease [41,42], with gastrointestinal and orthopedic comorbidities being reported in over 80% of individuals [43]. In general, the *IQSEC2* phenotypic spectrum has overlapping clinical features with RTT, Angelman syndrome, Pitt–Hopkins syndrome [10,11,18,24,25,44], and the *HNRNPH2*-related NDD [45]. Therefore, different genetic conditions with different underlying molecular mechanisms can share similar RTT-like phenotypes. Accordingly, some *IQSEC2* patients have had *MECP2* analysis, prior to the identification of the *IQSEC2* variant [11]. Recently, the provocative hypothesis of *IQSEC2*-related encephalopathy as a phenotypical extension of the “RTT spectrum continuum” has been raised [11].

Several gene variants can mimic RTT at the phenotypic level, with at least 66 different genes being so far reported in the medical literature [24,25,44,45]. Understanding the underlying causes of convergence of different NDDs towards an RTT-like phenotype can be of help in underpinning the genotype–phenotype basis of NDDs and lead to a better differential diagnosis and an early genetic screening.

The aim of this study was to obtain systematic information on natural history and extra-central nervous system (CNS) manifestations in over 90% of the Italian *IQSEC2* population using semi-structured family interviews.

## 2. Materials and Methods

### 2.1. Study Design and Ethics

The study was an observational interview-based study. We collected clinical and molecular data of a total of *n* = 19 participants (*n* = 10 females and *n* = 9 males) that were followed by different Italian clinical centers. The Italian *IQSEC2* parents’ association [46] facilitated the recruitment of participants that were willing to be interviewed for our study. Parents/caregivers were subsequently invited by telephone to participate. Upon acceptance of invitation, semi-structured video interviews were arranged and a written consent form was obtained by all the patient’s parents/legal guardians prior to the interviews. To the best of our knowledge, at least two more cases are known, although written consent was not obtained due to the caregivers/parents’ decision. All sensitive clinical data were anonymized by assigning a randomly generated integer code to each *IQSEC2* patient. Ethical approval was granted by the Ethics Committee of University Hospital (Azienda Ospedaliera Universitaria Senese) (Approval n. 4/2023 and approval date 4 May 2023).

An age-matched female RTT population (*n* = 10) with a clinical diagnosis of typical RTT and proven *MECP2* gene variant was selected from our internal clinical database (Azienda Ospedaliera Universitaria Senese, Siena, Italy) as a comparison group. Clinical and molecular data regarding *n* = 5 patients harboring pathogenic *IQSEC2* variants have been previously described [11,47].

### 2.2. Procedures

Interviews were conducted by an expert clinician in RTT and RTT-like conditions (C.D.F.) using telephone or videoconference facilities and were transcribed verbatim. The duration of each interview was approximately 1.5 h. The semi-structured interview was comprised of 4 different sections: (1) demographic/anthropometric information of *IQSEC2*-mutated subjects; (2) pre-/perinatal medical history and developmental milestones of subject; (3) clinical features of *IQSEC2* disease and presence of gastrointestinal disorders (GID); (4) quality of life. Whenever needed, the data obtained by interview were compared with the individual pediatric health book, clinical records, and laboratory data, mainly including genetic reports. In the lack of a clear description of clinical records, the main cranio-facial features were assessed on facial pictures kindly provided by parents/caregivers. In order to record specific information for comparison with RTT group and reduce potential bias, the same clinician (C.D.F.), after reviewing the patient information, also rated specific scales and questionnaires for clinical severity and quality of life (see details in Section 2.4 and Section 2.5) during the interview.

### 2.3. Study Variables

To filter and categorize the gene variants, we used the American College of Medical Genetics and Genomics (ACMG) criteria [48] and ClinVar database [49] for data validation (D.L.). Variants were classified according to the ACMG standards including five categories of pathogenicity: pathogenic, likely pathogenic, variant of uncertain significance (VUS), likely benign, and benign.

For demographic and anthropometric variables, the following information was considered: age, age at the time of the genetic diagnosis, side dominance, body weight z-score, height z-score, head circumference z-score and body mass index (BMI) z-score. For the *IQSEC2*-mutated patients, corresponding z-scores for body weight, height, head circumference, and BMI were calculated on the basis of physiological growth charts [50]. For the RTT patients, although validated RTT-specific growth charts [51] are known, for the sake of comparability with *IQSEC2* patients, the reference to standard growth charts was made [50]. Head control, sitting, and walking milestones were analyzed as early developmental milestones. For the section regarding pre- and perinatal history, we examined gestational age, prematurity, presence of adverse perinatal events, birth weight and its gestational age categorization (i.e., appropriate, large, or small for gestational age newborns), delivery mode, and breastfeeding and its duration. Clinical features of IQSEC2-related encephalopathy were subdivided into neurological (i.e., psychomotor regression, microcephaly, strabismus, intellectual disability degree, verbal language, ataxic gait, muscle hypotonia, hypoalgesia, autonomic nervous system (ANS) dysfunction, hand stereotypies, hand stereotypies pattern), orofacial (i.e., plagiocephaly, macrotia, micrognathia, bruxism, drooling), behavioral features (i.e., ASD-like features, mood changes, anxiety, crying/laughing), respiratory features (i.e., apneas, hyperventilation, air swallowing), and others (i.e., scoliosis, tendon retractions).

For epilepsy, we considered relevant information as the following items: epilepsy history, age of seizure onset, active epilepsy, seizure frequency, and antiepileptic drugs (AED). For gastrointestinal disorders (GID), we examined the co-presence of one or more among the following symptoms: dysphagia, gastrointestinal reflux, severe constipation, biliary disease, and presence of gastrostomy. GID severity was rated according to the number of elements. As potential interfering factor with GID, we considered the use of proton pump inhibitors (PPIs), laxatives, AED, antipsychotic drugs, non-steroidal anti-inflammatory drugs (NSAIDs), and steroidal anti-inflammatory drugs (SAIDs). Phenotypical abnormalities were presented by order of frequency of occurrence in the surveyed *IQSEC2* patients’ population, i.e., always present (100%), very frequent (90–80%), frequent (79–30%), occasional (29–5%), and rare (<5%) [52].

### 2.4. Clinical Severity

#### 2.4.1. Comparison *IQSEC2*-Related Encephalopathy with RTT

In order to discriminate overlaps/differences and to gauge the percentage of likelihood with RTT, the proportion of *IQSEC2* subjects who meet Neuls’ major and minor diagnostic clinical criteria for RTT were calculated and clinical severity was assessed using some international validated clinical severity scales for RTT and compared to an age-matched female RTT group (*n* = 10). The Rett Clinical Severity Score (CSS) [53] is a specific Likert scale designed to assess the natural history of key symptoms (i.e., age of onset of regression, somatic growth, head growth, independent sitting, ambulation, hand use, scoliosis, language, non-verbal communication, respiratory dysfunction, autonomic symptoms, onset of stereotypies, and seizures). The Motor Behavior Assessment Scale [54] is designed to survey movement abnormalities, especially extrapyramidal symptoms, behavioral problems, and abnormal physiological features in individuals with RTT. It is a Likert checklist of 37 items subdivided in social, communication skills and adaptive behaviors (MBAS I), orofacial and respiratory abilities (MBAS II), and motor abilities/physical signs (MBAS III). The Rett Syndrome Behavior Scale (RSBQ) [55], a Likert checklist of 45 items, measures behavioral and emotional features as well as movement abnormalities. RSBQ consists of 8 subdomains: general mood, breathing, hand behavior, repetitive face movements, body rocking and expressionless face, night-time behavior, fear/anxiety, and walking/standing. The Gastrointestinal Health Questionnaire (GHQ) is a clinical outcomes measure of gastrointestinal health in RTT based in gastrointestinal health, function, medication use, and surgical interventions [56]. Stool form and consistency were assessed by the Bristol Stool Form Scale, a 7-point scale extensively used in clinical practice [57].

#### 2.4.2. Genotype–Phenotype relationships

In order to test a possible genotype–phenotype relationships, illness severity was compared as function of type of *IQSEC2* variant (i.e., missense, in-frame deletion, frameshift, splicing, nonsense). We combined *IQSEC2* frameshift and nonsense variants as null variants theoretically leading to a more severe phenotype. Null variants were then compared with all the other *IQSEC2* variants.

#### 2.4.3. Comparison of *IQSEC2* as a Function of Initial RTT Diagnosis vs. RTT

In order to discriminate overlaps /differences between the female *IQSEC2* patients initially diagnosed as RTT (*n* = 4) and RTT patients (*n* = 10), illness severity was compared for the following groups: *IQSEC2* females not initially diagnosed as RTT, *IQSEC2* females initially diagnosed as RTT, and RTT patients harboring *MECP2* variant and presenting Neul’s criteria for typical disease. Likewise, the same markers described in Section 2.4.1 were assessed in these groups.

### 2.5. Sleep Quality

Sleep quality was evaluated by the Sleep Disturbance Scale for Children Questionnaire (SDSC) [58], one of the most used assessment tools for pediatric sleep. The questionnaire consists of 26 items grouped in 6 subscales relating to the major sleep complaints in pediatric age: disorders in initiating and maintaining sleep (DIMS), sleep breathing disorders (SBD), disorders of arousals/nightmares (DA), sleep/wake transition disorders (SWTD), disorders of excessive somnolence (DOES), and sleep hyperhidrosis (SHY). Total sleep time and sleep onset latency were classified according to [58]. Co-sleeping behavior frequency of patients was explored. Furthermore, sleep quality was assessed as a function of initial diagnosis of RTT.

### 2.6. Quality of Life

The Quality of Life Inventory-Disability (QI-Disability) is a 32-item questionnaire assessing the quality of life of children with ID [59,60]. The questionnaire is comprised of six domains: social interaction (7 items), positive emotions (4 items), negative emotions (7 items), physical health (4 items), leisure and the outdoors (5 items), and independence (5 items).

Moreover, the quality of life of the *IQSEC2* females was assessed as a function of initial diagnosis of RTT.

### 2.7. Statistical Analysis

All variables were tested for normal distribution (D’Agostino Pearson test), and data were presented as means ± standard deviation or medians and interquartile range for continuous normally distributed and non-Gaussian variables, respectively. The differences between RTT and control groups were evaluated by independent-sample *t*-test (continuous normally distributed data), Mann–Whitney rank sum test (continuous non-normally distributed data), chi-square statistics (categorical variables with minimum number of cases per cell ≥ 5), or Fisher’s test (categorical variables with minimum number of cases per cell 0.5 was accepted to indicate good discrimination). Relationships between variables in univariate analysis were tested by linear regression analysis or Spearman’s rank correlation. Analysis of variance was performed by one-way ANOVA or Kruskal–Wallis test, as appropriate. Comparisons against a randomly generated simulated control group for menarche age, based on an Italian secular trend survey [61] was performed. To identify the cut-off value for age at menarche in the IQSEC2-mutated females vs. simulated control group, a receiver operating characteristic (ROC) curve analysis was performed. To handle the missing data, the technique of simple omission was used [62]. The MedCalc version 20.013 statistical software package (MedCalc Software Ltd., Ostend, Belgium; https://www.medcalc.org; 2021) was used for data analysis, and a two-tailed *p* < 0.05 was considered to indicate statistical significance.

## 3. Results

### 3.1. Prevalence Estimates of IQSEC2-Encephalopathy in Italy

For the first time, by gathering the clinical statistics of >90% of the recognized national *IQSEC2* cases, we were able to infer the possible prevalence estimate for the disorder within the Italian territory. As a function of same-age class and gender in Italian population [63], the prevalence estimates were calculated (Appendix A). In particular, prevalence of *IQSEC2* disease resulted in female subjects 0.0000702/100,000 (95%C.I. 0.0000674–0.0000719) and 0.0000786/100,000 (95%C.I. 0.0000714–0.000858) in male subjects, respectively.

With regards to RTT, unfortunately, the precise prevalence of RTT in Italy is still unclear. The most accurate figures of prevalence and incidence have been estimated at 3.2 and 2.3/100,000, at least in the US [64], although a recent systematic review reported a pooled prevalence estimate of 7.1/100,000 females (95% CI: 4.8, 10.5) in the general female population [65].

### 3.2. Molecular Data of the IQSEC2 Population

The *IQSEC2* pathogenic variants identified in the Italian population were categorized as: *n* = 12/19 (63.1%) de novo variants, *n* = 5/19 (26.3%) maternally inherited, and *n* = 2/19 (10.5%) unknown (i.e., mother’s data were unavailable for Patient ID #3, whereas genetic analysis for the parents of Patient ID #18 was in progress) (Figure 1). Type of *IQSEC2* mutations included frameshift (*n* = 6), splicing (*n* = 4), missense (*n* = 4), in-frame deletion (*n* = 1), and nonsense (*n* = 4). Overall, a total of *n* = 10 *IQSEC2* patients harbored null variants (*n* = 4 males and *n* = 6 females). Of interest, *n* = 6 *IQSEC2* patients (*n* = 4 males and *n* = 2 females) harbored variants in other genes (the list of these genes is reported in Appendix A).

### 3.3. IQSEC2-Related Encephalopathy General Overview

Demographic, biometric, and clinical features of the examined *IQSEC2* patients are illustrated in Table 1. Mean patients’ age was 13.7 ± 7.4 years (pediatric patients, *n* = 9, 47.4%, adolescents, *n* = 6, 31.6%, and adults, *n* = 4, 21.1%; *p* = 0.094). The hallmark features already described for the syndrome [5,10] were confirmed in our examined case series. None of them, however, was constantly present, with the single exception of ID if also moderate (*n* = 3, 15.8%) and mild (*n* = 1, 5.3%) forms are included. Developmental delay and seizures were very frequent signs, i.e., detectable in the 80–99% prevalence range. ASD-like features and severe ID were frequent, as well as muscle hypotonia (i.e., 78% in affected males and 40% in females, respectively). In contrast, developmental regression was an occasional report, falling in the 5% to 29% sign frequency range. As it concerns the presence of other possibly diagnostic signs, the parents reported a series of signs/symptoms in descending order of frequency. The most frequent one (84.8%) regarded a decreased pain sensitivity/increased pain threshold. Hand stereotypies and sleep disturbance were reported in 68.4% of patients; autonomic dysfunction and drooling 63.2%; hand use deficit, mood changes, uncontrolled seizures, and self-aggressiveness 57.9%. Again, as frequent features, the following signs were reported: anxiety (52.6%), mandatory co-sleeping and ataxic gait (51.4%), unexplained laughter episodes (47.4%), microcephaly, scoliosis and strabismus (42.1%), cranio-facial dysmorphisms and severe constipation (36.8%), along with ADHD-like features and combined deficit in walking and hand use (31.6%). As occasional findings, somatic growth deficit (21.1%) and febrile seizures (13.3%) were also recorded.

### 3.4. Perinatal/Neonatal Data, Key Early Developmental Milestones, and Clinical Features as a Function of Gender in the IQSEC2 Population

No significant differences were observed between *IQSEC2* males and females. An early genetic diagnosis (i.e., ≤2 years) was achieved only in *n* = 1/19 *IQSEC2* patients (i.e., 5.3% of total, male subject). For anthropometric variables, no statistical differences were observed for height z-score, head circumference z-score, and BMI z-score, with the single exception of body weight z-score, which resulted in being significantly reduced in *IQSEC2* male in contrast to that of the *IQSEC2* female group (*p* = 0.040). *IQSEC2* patients showed on average a physiological perinatal course and neonatal data (Table 2). Adverse perinatal events were reported in 31.6% of the *IQSEC2* population, while no statistically significant differences were observed as a function of gender. No difference was observed regarding the frequency of continuous breastfeeding (at 12 months) (*p* > 0.984). Exclusive breastfeeding was reported in 57.9% of the *IQSEC2* patients, with a trend towards females (70%) as compared to males (44.4%) although the difference is not statistically significant (*p* = 0.273). For early developmental milestones, head control was achieved late in 37.5% of the *IQSEC2* subjects as well as autonomous sitting (63.2%) (Table 1). There were no statistical differences in early gross motor milestone acquisition as a function of gender that was observed. For autonomous walking, only 10.5% of *IQSEC2* patients achieved the milestone on time, while 57.9% acquired this functional skill late as compared to the physiological range, 5.3% lost this skill, and 26.3% did not achieve it. For this milestone, a different pattern was found between affected males and females. In particular, only *n* = 2 females reached this milestone within the physiological range. No males appear to acquire this functional step in the physiological time window. Over the same period, three affected males (33.3%) achieved this skill late, one of them (11.1%) has since lost this skill, and the remaining five patients never acquired autonomous walking (Table 1). The pattern of autonomous walking acquisition significantly differs in females, in that 20% acquire this functional skill within the physiological time window while 80% of them achieve this step late (*p* = 0.017). The timing of both autonomous sitting and autonomous walking milestones appears to be altered in *IQSEC2*-mutated patients. In particular, when compared to physiological range, the autonomous sitting acquisition was delayed in 63.2% of the patients, with a significant trend in male patients (*p* = 0.005). For the autonomous walking, only 10.5% of *IQSEC2* patients acquired the milestone in the physiological range. We observed significant differences both in male (*p* = 0.0002) and female patients (*p* = 0.0022). For head control milestone, a trend was observed for affected males (*p* = 0.072). Although speech was either delayed or regressed in the great majority of *IQSEC2* patients (>80%), there were no statistical meaningful differences between genders and/or groups that were observed (*p* ≥ 0.606). Likewise, no statistical differences were observed for side dominance (data not shown, *p* = 0.988) (Table 1).

For the neurological features, psychomotor regression was present in 15.8% of the *IQSEC2* population, with a trend towards *IQSEC2* male patients (33.3%) as compared to the female group (0%) (*p* = 0.087) (Figure 2). Furthermore, albeit with different intensity, ID was reported for all patients. Aphasia and strabismus were the most prevalent neurological features in the *IQSEC2* male patients as compared to the affected female population (66.7% vs. 20%, respectively, *p* = 0.045). Conversely, ANS dysfunction was more prevalent in *IQSEC2* females as compared to that of *IQSEC2* males (90% vs. 33.3%, *p* = 0.013). No further difference between genders reached statistical significance. Furthermore, no statistically significant differences observed for orofacial, behavioral, respiratory, and miscellaneous features (*p* > 0.165), with the single exception of hyperventilation, which was reported to be more prevalent in *IQSEC2* females as compared to affected males (40% vs. 0%, *p* = 0.038). Interestingly, obstructive sleep apnea syndrome (OSAs) was reported in *n* = 4 female *IQSEC2*-mutated patients.

Combined autonomous walking and purposeful hand use skills were conserved in *IQSEC2*-mutated female patients as compared to affected males (lost skills in 55.6% vs. 10%, *p* = 0.038).

### 3.5. IQSEC2-Related Encephalopathy vs. RTT: Overlaps and Differences

#### Clinical Course

Age and age category at the time of the interview were not significantly different between *IQSEC2* patients and typical RTT patients (*p* ≥ 0.210) (Table 2). Early genetic diagnosis (≤2 years) was considerably different between *IQSEC2* patients and RTT girls (*p* = 0.0005). Likewise, anthropometric measures were notably spared in the *IQSEC2* females as compared to RTT of the same gender (*p* ≤ 0.015), although a non-significant trend was observed for BMI z-score (*p* = 0.062). Perinatal and neonatal history was uneventful, with the single exception of breastfeeding frequency (Table 2) in that the male *IQSEC2* group presented a reduced frequency (44.4%) as compared to the other two groups (70% in female *IQSEC2*: 70%, RTT: 100%, *p* = 0.025, Chi-squared test for trend *p* = 0.0067).

Early gross motor developmental milestones were significantly delayed in the *IQSEC2* patients as compared to RTT (*p* ≤ 0.033) whereas the acquisition of verbal language was similarly either delayed or regressed between two groups (*p* = 0.338) (Figure 3). No statistically significant differences were observed for side dominance (data not shown, *p* = 0.698).

Psychomotor regression was significantly less frequent in *IQSEC2* patients as compared to the RTT subjects (*p* < 0.0001), as well as aphasia (*p* = 0.001) (Figure 4). An increased frequency of strabismus was reported in *IQSEC2* males (*p* = 0.004). Interestingly, frequency and pattern of hand stereotypies significantly differed between groups: for *IQSEC2* patients, a lower frequency of stereotypies was reported in the female group (50%) (*p* = 0.015), while the hand stereotypie patterns were primarily related with separated hands in the *IQSEC2* patients vs. midline stereotypies in the RTT population (*p* = 0.0071, chi-squared test for trend, *p* = 0.0083). Onset of hand stereotypies also differed between RTT and female *IQSEC2* patients, in that around 60% RTT had onset <18 months of life vs. 28.6% female *IQSEC2*. *IQSEC2*-mutated males showed more overlaps with RTT patients, in that stereotypies generally started before 18 months of life in 75% of patients. Frequency of ID degree, microcephaly, ataxic gait, muscle hypotonia, diminished pain response, and ASN dysfunction did not significantly differ between groups, (*p* ≥ 0.084). ASD-like behavioral features were significantly more represented in the *IQSEC2* patients’ population than in RTT (*p* = 0.0008, chi-squared test for trend, *p* = 0.0006), whereas the frequency for the other examined behavioral problems did not significantly differ between groups (*p* ≥ 0.316). Reported frequency of respiratory symptoms was significantly lower in *IQSEC2*-related encephalopathy as compared to RTT patients (*p* ≤ 0.025). On the other hand, no significant differences were observed for the orofacial features (*p* ≥ 0.123) or miscellaneous signs (*p* ≥ 0.174). A positive history for epilepsy was present in 100% of *IQSEC2* male patients as compared to 90% for both *IQSEC2* female and RTT patients, with no statistical differences between groups (*p* = 0.967) (Figure 5). Although uncontrolled epilepsy was seemingly more prevalent in *IQSEC2* males as compared to *IQSEC2* females, the difference was not substantial (*p* = 0.248). The gap between developmental delay signs and epilepsy onset is 2.45 ± 2.7 (0.4–4.5 years) years in *IQSEC2* males and 1.8 ± 1.8 (0.3–3.3 years) in *IQSEC2* females as compared to that observed in RTT patients: 2.6 ± 2.2 years (0.7–4.5 years).

The prevalence of febrile seizure was also investigated. No significant difference was observed between groups (*IQSEC2* males: 0%, *IQSEC2* females: 25% and RTT: 20%, *p* = 0.380). Information was unavailable for 4 patients (*n* = 2 males, *n* = 2 females). Examined GID symptoms included constipation, dysphagia, and gastrointestinal reflux disease (GERD). The only significant difference was a lower frequency of constipation in the *IQSEC2* group (*p* = 0.002), whereas the prevalence of dysphagia and GERD was comparable between groups (*p* ≥ 0.077). Antipsychotic drugs were administered in *n* = 1 male *IQSEC2*-mutated patients, *n* = 2 female *IQSEC2*-mutated patients, and *n* = 3 *MECP2*-mutated patients (*p* = 0.596).

Although no difference was observed for stool consistency (i.e., Bristol stool type) between the examined groups (Figure 5), it is interesting to note that the majority of male *IQSEC2*-mutated patients (66.7%) apparently showed a Bristol stool type corresponding to constipation, likely due to reduced use of the laxatives. None of *IQSEC2*-mutated patients and *MECP2*-mutated patients took anti-inflammatory (NSAID/SAID) therapy. Intestinal dysbiosis and inflammation are known features already reported in RTT [66]. Interestingly, intestinal dysbiosis was reported in *n* = 1 female *IQSEC2*-mutated patient (Patient ID #6) and intestinal inflammation in *n* = 1 female *IQSEC2*-mutated patient (Patient ID #9), respectively.

### 3.6. Illness Severity

Overall, illness severity, as rated by RTT-specific tools, was found to be lower in the *IQSEC2*-related encephalopathy as compared to typical RTT patients (Figure 6) with lower total CSS and CSS subdomains (*p* values range 0.037 to 0.0001). Of note, total CSS, CSS onset, and CSS Behav/Neuro/Comm were significantly lower in the *IQSEC2*-mutated females as compared to *IQSEC2* males and RTT patients (Figure 6). Likewise, total MBAS score, but not MBAS I (*p* = 0.346), as well as MBAS II (*p* = 0.028) and MBAS III (*p* < 0.001) subscales were significantly lower in the *IQSEC2* patients’ group. RSBQ and its sub-areas showed significant differences between *IQSEC2* and RTT (*p* ≤ 0.029) with the exceptions of general mood, fear/anxiety, and repetitive movements sub-scores (*p* ≥ 0.354).

### 3.7. Sleep Disturbance

In the *IQSEC2*-mutated patients, sleep quality, as rated by specifically validated family questionnaires, appears to be consistently disrupted (Figure 7). Sleep disturbance (i.e., SDSC total score >39) was evidenced in the majority of the *IQSEC2*-mutated male population (7/9 = 77.8%) and in slightly more than half of the affected females (6/10 = 60%). All the SDSC sub-areas were comparable between groups (*p* ≥ 0.147). The prevalence of altered sleep in *IQSEC2* was comparable to the degree of sleep disturbance (i.e., 80%) observed in the typical RTT patients (difference between groups, *p* = 0.5518). Mean total sleep time (hours) was similar between groups: *IQSEC2* males, 8.5 [8.5–10], *IQSEC2* females, 10 [10–10], RTT, 10 [7.5–10], *p* = 0.279), as well as sleep onset latency (min) (*p* = 0.4681, data not shown).

Administration of sleep-influencing drugs (i.e., melatonin, hydroxyzine, niaprazine) was reported in 20% of female *IQSEC2*-mutated patients (*n* = 2), and 20% RTT patients (*n* = 2) and none of the male *IQSEC2*-mutated patients, (*p* = 0.035). The effect of these drugs on the SDSC total score and its sub-scores was tested. No significant difference was found for *IQSEC2*-mutated patients (*n* = 2 treated patients 47.5 ± 10.6 vs. 40.5 ± 9.8 untreated patients, *p* = 0.397), whereas RTT patients treated (*n* = 2) exhibited a higher SDSC total score as compared to untreated RTT subjects (53.5 ± 4.9 vs. 41.5 ± 5.3, *p* = 0.020). For female *IQSEC2*-mutated patients, no significant difference was observed for DIMS, DA, SWTD, and DOES sub-scores (*p* ≥ 0.405), as well as total sleep time and sleep onset latency (*p* ≥ 0.552), while SBD and SHY sub-scores resulted increased as a function of sleep-influencing drugs (*p* = 0.003 and *p* = 0.035, respectively). In the examined RTT group, the treatment induced an increase in DA, SWTD, and DOES sub-scores (*p* = 0.035, *p* = 0.030, *p* = 0.004) and a decrease in SHY sub-score (*p* = 0.021). No statistically significant difference was observed in DIMS, SBD, total sleep time, and sleep onset latency (*p* ≥ 0.287).

### 3.8. Quality of Life

As the last but not certainly the least relevant aspect in each interview, the quality of life was assessed by the Quality of Life Inventory-Disability (QI-Disability) (Figure 7). A non-statistically significant trend was observed for total QI-Disability and family/friends interaction section in *IQSEC2*-encephalopathy (*p* = 0.070), suggesting the impact of the disease on the quality of life of subjects and social interactions with both family components and friends. *IQSEC2* patients with sleep disturbance (i.e., SDSC total score > 39) (*n* = 13/19) showed a higher QI-Disability score (median 79, IQR 71.7–92.2) than patients without relevant sleep problems (median 67.5, IQR 67–77) (*p* = 0.044). Interestingly, we observed a significant difference for the feeling emotions section (*p* = 0.014) in *IQSEC2* patients as compared to RTT patients, confirming the presence of behavioral components in the *IQSEC2* patients, including aggressiveness, mood changes, bruxism, or crying. From the interview, *n* = 15/19 (78.9%) *IQSEC2*-mutated patients were followed by either child neuropsychiatrists or neuropediatricians. As it concerns the other professionals taking care of the complex needs of these patients, the landscape is remarkably more scattered in that about half of *IQSEC2* patients lack a multi-specialistic support team (data not shown).

### 3.9. Miscellaneous Findings

Other relevant clinical features were reported for some peculiar *IQSEC2*-mutated patients. In particular, the parents/caregivers reported daily bradycardia episodes along with the co-presence of aortic insufficiency, likely due to a bicuspid valve in one *IQSEC2*-mutated male patient (Patient ID #13). Actually, this patient was found to harbor a variant in the *TGFBR1* gene linked to Loeys-Dietz 1 syndrome and the presence of a bicuspid valve (Appendix A). In another *IQSEC2*-mutated male patient, three episodes of brief and resolved unexplained events (BRUE) [67] were reported at the age of 5 months, along with bradycardia (Patient ID #18). Osteopenia was reported in one *IQSEC2*-mutated female patient (Patient ID #8), although, unfortunately, bone density parameters are not routinely assessed in *IQSEC2*-related encephalopathy patients, with the individual exception of cases initially misdiagnosed as RTT.

### 3.10. IQSEC2 Genotype–Phenotype Relationships

Individual data on illness severity are reported in Appendix A. No statistically significant genotype–phenotype relationship using the examined severity markers (i.e., CSS, MBAS, and RSBQ) was observed in the *IQSEC2* population. No difference was observed nor in sleep disturbance scale (SDSC) or QI-disability inventory (QI-Dis).

In order to test the possible impact of maternal age at conception/delivery on de novo *IQSEC2* mutations, we compared age at delivery of mothers with that of matched controls from national institutional statistical reports [68]. Interestingly, while the age of mothers at delivery for *IQSEC2* inherited mutation (31.6 ± 2.6 yrs) did not differ from delivery year matched controls (30.7 ± 0.4 yrs, *p* = 0.441), the age of the mother with offspring carrying the *IQSEC2* de novo mutation was significantly older than control mother (33.1 ± 3.6 yrs, *p* = 0.031). Since several reports suggested advanced paternal age on generating carrying gene mutation causing NDDs in offspring [69,70], we also tested the comparison of paternal age with a control paternal link [71]. No difference was observed for paternal age, either for de novo variants or inherited variants (*p* = 0.426 de novo and *p* = 0.929 inherited variants).

### 3.11. Novel Findings in IQSEC2-Related Encephalopathy

#### 3.11.1. Age at Menarche

Although no cases of precocious puberty were reported, a significantly earlier age at menarche was observed for the *IQSEC2* female group from our examined Italian case series (10.7 ± 1.4 years, range 9.0 to 12.5 years, 95% C.I.: 9.7 to 11.7 years). From an ROC curve analysis, a cut-off age at menarche ≤ 12.5 years significantly discriminated the female *IQSEC2* patients from the age at menarche of a randomly generated simulated control group based on an Italian secular trend survey [61] (AUC 0.889, SE 0.075, 95% CI AUC 0.661 to 0.995, *p* < 0.001). Comparisons against the simulated control group for menarche age yielded *p* < 0.05 in 11 out of 20 tests (*p*-value range 0.007 to 0.04) against non-significantly statistical comparisons in the remaining 9 tests (*p*-value range 0.057 to 0.250) (Figure 8). Although age at menarche was significantly related to age of the mother only in the RTT population (r = 0.720, *p* = 0.05, *n* = 16), interestingly, no correlation between maternal menarche age and menarche age of female *IQSEC2*-mutated patients was observed (r = 0.241, *p* = 0.568, *n* = 8).

#### 3.11.2. Mandatory Co-Sleeping Behavior

Mandatory (i.e., every night) co-sleeping behavior was reported in 50% of the *IQSEC2* females, none of the *IQSEC2* males, and 10% of RTT patients (*p* = 0.0159) (Figure 9). Severe co-sleeping behavior among the *IQSEC2* females was unrelated to severity of sleep disorder, total sleep time, sleep onset latency, psychomotor regression, and autistic-like traits (*p* ≥ 0.180, data not shown). However, a non-statistically significant trend in increased motor severity was observed (*p* = 0.078). Mandatory co-sleeping behavior in *IQSEC2* was related to female gender, de novo mutation origin, younger age, taller height with underweight, better social interaction, and low life quality impact for the family and friends’ area (Figure 9). Mandatory co-sleeping was not related to AED therapy (*p* = 0.548).

#### 3.11.3. Behavioral and Sleep Features in *IQSEC2* Null Variants

We tried to understand whether *IQSEC2* null (frameshift + nonsense) variants might be differentiated phenotypically from other variants. Interestingly, null variants were responsible for distinct behavioral features in *IQSEC2*-mutated patients. Indeed, RSBQ total score, RSBQ general moods, and RSBQ for non-categorized items are statistically higher than those assessed in patients harboring other *IQSEC2* variants (*p* = 0.012, *p* = 0.006, and *p* = 0.010, respectively) (Figure 10A). It should be reminded that the RSBQ non-categorized subdomain includes consciousness, behavioral, and neurological symptoms.

Patients harboring *IQSEC2* null variants showed a significantly high rate of sleep disturbances (i.e., SDSC > 39): *n* = 8/10 (80%) vs. other *IQSEC2* variants *n* = 2/9 (22.2%) (*p* = 0.023 Fisher’s exact test). In particular, the ability in initiating and maintaining sleep was more impaired in patients with null *IQSEC2* variants (*p* = 0.039) (Figure 10B), along with a higher impact on the health wellbeing sub-item (*p* = 0.042) of the QI-Disability questionnaire. As it concerns autonomous walking and purposeful hand use, both skills tend to be more conserved in patients harboring other *IQSEC2* variants (88.9%) as compared to patients harboring *IQSEC2* null variants (50%), although the difference was not statistically significant (*p* = 0.141).

### 3.12. Diagnostic Criteria for RTT in IQSEC2-Related Encephalopathy

In the actual natural course, only *n* = 5 patients (four of whom being girls) were initially suspected as having RTT (Patient ID#4, #8, #11, #14, and #16) (Table 3). Interestingly, however, only for one of them was psychomotor regression reported. Before genetic diagnosis for *IQSEC2*, descriptive diagnosis before the genetic diagnosis of *IQSEC2*-related encephalopathy was reported for *n* = 7 patients (#2, #5, #6, #10, #12, #13, #17), whereas a psychiatric disorder of unlikely genetic origin was suspected for *n* = 3 patients (#3, #9, #19). In one case, a GLUT1 deficiency syndrome vs. Lennox–Gastaut syndrome (Patient ID #18) was suspected (Table 3). On the other hand, Angelman syndrome was suspected for *n* = 3 [#7, #11, and #15 of whom one also suspected as RTT (#11)]. The clinical phenotype of one patient was supposed to be linked to an inborn error of metabolism vs. mitochondrial disease (#1). By the way, this patient was proven to harbor a variant in the *SMARCC1* gene (Appendix A), one of the 66 genes reported to be linked to the RTT-like phenotype along with *IQSEC2* gene pathogenic variant [44]. Interestingly, this patient showed a severe somatic growth deficiency (BW z-score −2.33 vs. −0.07 ± 1.3 no-*SMARCC1* co-mutated *IQSEC2* patients, height z-score −1.88 vs. 0.05 ± 1.1, BMI z-score −2.33 vs. −0.17 ± 1.42, and head circumference z-score −2.05 vs. −0.86 ± 1.02). Although no particular differences emerged regarding clinical severity, perinatal history, hallmark features, and developmental milestones, hand stereotypies onset was earlier (below 18 months) than other *IQSEC2* patients and the average of total sleep time was shorter (7.5 h vs. 9.2 ±1.3 h). In a post hoc analysis, a total of 8/19 (42.1%) *IQSEC2*-mutated patients showed all of the four major criteria (*n* = 7 males and *n* = 1 female), and *n* = 6 showed supportive criteria for atypical RTT (*n* = 1 male and *n* = 5 females) (Table 3). However, a regression followed by recovery or stabilization period was reported only in *n* = 4 cases, of whom *n* = 3 cases with four major criteria and *n* = 1 case with supportive criteria (Table 3). We explored, interestingly, overlaps and discrepancies between *IQSEC2* patients initially diagnosed as RTT or RTT-like. The *IQSEC2* females not initially diagnosed as RTT did not differ in terms of mutation severity class from the *IQSEC2* females not diagnosed as RTT (*p* ≥ 0.153). In particular, *IQSEC2* females initially diagnosed as RTT appear to be either an isolate group (total CSS) or closer to one of the remaining groups, either *IQSEC2*-mutated females not diagnosed as RTT (CSS onset, MBAS III, SDSC-SBD, GHQ, 32-Dis Feeling-Emotions, ASD-like behavior, microcephaly, non-midline hand stereotypies, walking + hand use) or RTT (CSS Behav/Neuro/Commun, language regression, AED therapy, OSAS) (Figure 11).

## 4. Discussion

Research on the clinical course of Italian *IQSEC2*-related encephalopathy has been incompletely investigated thus far [11]. In the present study, systematic data regarding the natural course of the condition including early developmental history, epilepsy onset, sleep disturbances, and GID were obtained by structured family interviews, along with semi-quantitative data on illness severity, sleep quality, behavioral changes, and their impact on the quality of life. A systematic assessment of gender-related differences and genotype–phenotype relationships was obtained. Finally, the medical history and the clinical features of *IQSEC2*-mutated females were compared with those of a population of typical RTT female patients with a similar age distribution, in order to further test the intriguing hypothesis of a “RTT spectrum continuum” as previously raised by Lopergolo et al. [11]. The first question we tried to address was the prevalence estimate for the *IQSEC2*-related encephalopathy in Italy. To date, the number of patients diagnosed worldwide is in the order of a hundred cases [5,10,11,12,17,18,19,20,21], although figures regarding the incidence and prevalence of the disease are unclear to date. For the first time, by gathering the clinical statistics of all the recognized *IQSEC2* cases in Italy, we decipher the possible prevalence within the Italian territory, which was found to be in the order of 0.0000702/100,000 females and 0.0000786/100,000 males, respectively. It should be, however, noted that this prevalence is certainly underestimated, since it does not account for at least two further Italian patients harboring an *IQSEC2* gene mutation for whom informed consent was not obtained and therefore clinical data are unknown to us (see Methods). Nevertheless, from this rough estimate, we could infer that *IQSEC2*-related encephalopathy is very likely to be highly underdiagnosed in our country if compared to the reported frequency of *IQSEC2* pathogenic variants in 2% of all patients with ID and epilepsy who underwent a whole exome sequencing analysis [15]. As it concerns RTT, although the precise prevalence of RTT in Italy is still unknown, the most accurate figures of prevalence and incidence of RTT in the US have been estimated at 3.2 and 0.23/100,000, respectively [64]. Nevertheless, a recent systematic review reports a pooled prevalence estimate of 7.1/100,000 females (95% CI: 4.8, 10.5) in the general female population [65]. Prevalence of *IQSEC2*-related encephalopathy in Italy seems to be at least two orders of magnitude lower than that of RTT. The condition is likely to be underdiagnosed when the current prevalence figures are compared to the observed frequency of *IQSEC2* pathogenic variants in 2% of all patients with ID and epilepsy who underwent a whole exome sequencing analysis [15]. From our survey, it appears that genetic diagnosis of the disorder is only occasionally achieved in infancy despite a clinical onset before the first 24 months of life.

In our examined *IQSEC2* population, none of the reported signs is to be considered as pathognomonic. However, a full host of diagnostic signs as previously reported in the literature were present, including the already described hallmark features according to [5,10]. None of these features was constantly present, with the single exception of ID. This feature has not been explored in experimental models, although whenever the cognitive function has been looked at [72,73], it was considered to be somewhat decreased. In contrast, autistic-like behavior and epileptic seizures were observed in all the mouse models recapitulating the disease [22,72,73,74,75]. Furthermore, decreased fertility, hyperactivity, defects in social interactions, and abnormalities in the electrophysiology of isolated neurons, with the single exception of anxiety reported only in two [73,75], have been reported. Meanwhile, in the present survey, the main aspects of the syndrome include developmental delay, seizures, ASD-like features, and muscle hypotonia; developmental regression instead was found as an occasional report. This point should be further explored in other *IQSEC2*-mutated populations. Interestingly, ASD-like features were previously reported in the literature but with a reduced frequency (i.e., 25% of all the *IQSEC2* described cases) [5]. Of course, further systematic ASD diagnostic investigation is needed before drawing a conclusion from these data in our survey. Furthermore, a number of ancillary signs were reported, including decreased pain sensitivity and/or an increased pain threshold. Abnormal pain sensitivity was a very frequent sign in the Italian *IQSEC2*-related encephalopathy patients. The *IQSEC2* gene is linked to the pain sensitivity issue [76], and interestingly, a single case report [20] with reduced pain sensitivity has been previously described. The reasons for this finding are still unclear, and given its apparently high frequency, certainly need further investigation. In this survey, strabismus (either as a transient, or more rarely, a permanent feature) was reported in slightly less than half of the examined patients. Among the ophthalmological features, strabismus has been already reported, although its cause/s (cortical blindness?) remain unclear [9,10,19]. Perinatal and neonatal history was essentially unremarkable. Nevertheless, as compared to the national frequencies of breastfeeding [77], a statistically significant difference was observed for the frequency of participants who exclusively breastfed at 4–5 months in the *IQSEC2*-mutated patients (57.9% *IQSEC2* patients vs. 23.6% national survey, *p* = 0.034). In contrast, no significant difference was observed for the frequency of continuous breastfeeding (at 12 months) (*p* > 0.984). Moreover, the latest Italian survey, between December 2018 and April 2019, remarks that less than one quarter (23.6%) of children aged 4–5 months were exclusively breastfed, presenting differences between geographical areas. An early developmental delay was reported in key gross motor developmental steps, such as independent sitting and independent walking with resulting gender differences with a significant developmental disadvantage for male patients, at least in the physiological age range of acquisition of autonomous walking. These data confirm the relevance of gender in the *IQSEC2* syndrome for gross motor development and motor skills. This is a key point in order to raise early clinical suspicion towards developmental encephalopathy. In our case series, we observed that the developmental delay almost always precedes the onset of epilepsy on average by 1.8 to 2.45 years in male and female patients, respectively. This is a key feature observed in developmental encephalopathies. The concept of developmental encephalopathies has progressively broadened with the discovery of a consistent fraction of the genetic basis of epilepsy [78,79,80]. In particular, this paradigm change historically coincides with the rise of the whole exome sequencing era [81]. An early genetic diagnosis of *IQSEC2*-encephalopathy at this stage would offer an opportunity for epilepsy prevention, thus theoretically offering a chance for modifying the natural course of the disease. The genotype–phenotype relationship for pathogenic variants in *IQSEC2* remains complex [12,13,14,18,44,82,83]. In that, it has been suggested that the phenotype is influenced by variants type, variants position, and patients’ gender [19], with the overall comparison of male and female phenotypes revealing a lower severity in females than males [12,13,83]. In our study, aphasia and strabismus appear to be more prevalent in the affected male population as compared to the female patients’ counterpart, while signs of dysfunction of the ANS appear to be more prevalent in the affected *IQSEC2* female population. As a possible co-morbidity, hyperventilation was reported significantly more frequently in affected females in our survey and needs more investigation. By using semi-quantitative tools, the disease appears to be substantially more severe in the affected *IQSEC2* males as compared to the female patient counterparts, at least as it concerns the history of disease onset, and the motor subdomains. Clinical data from the literature suggest that the disease is more severe in males than females [13]. In our survey, affected males showed lower body weight *z*-scores as compared to female patients. Somatic growth deficit in patients with pathogenic *IQSEC2* variants has been generically reported [10]. However, a gender difference has not been previously documented. De novo variants are genetically different from inherited variants, and are more damaging because mutation processes in de novo variants are happening between generations without undergoing purifying selection [84]. De novo, truncating variants correlate with severe disease in both female and male patients harboring an *IQSEC2* mutation [14]. Since *IQSEC2* is known to escape X inactivation in humans, female carriers can benefit from one normal *IQSEC2* allele. Paradoxically, however, the levels of *IQSEC2* are regulated in females such that the levels of *IQSEC2* are approximately the same in both genders [85], although at least one report indicates that *IQSEC2* is expressed at higher levels in males than in females in the brain cortex [86]. The reason for this discrepancy remains unclear, although the level of *IQSEC2* may be tightly regulated, at least in females. Inter-individual differences in this regulatory mechanism may explain why related females heterozygous for the same mutation show differing severity of the disease [13]. In our survey, males showed a higher rate of the not acquired walking milestone, along with a lower rate of motor skills (i.e., walking and hand use) as compared with affected females. Likewise, *IQSEC2* male patients exhibit a higher prevalence of ophthalmological signs (strabismus) and a higher clinical severity (CSS total score).

Theoretically, *IQSEC2* null (frameshift + nonsense) variants could be associated to a more severe phenotype. In our survey, patients with *IQSEC2* null variants showed more severe behavioral features and a higher frequency of sleep disturbances, thus suggesting a potential role of the *IQSEC2* protein in behavior and sleep modulation. In particular, for the behavioral features, the most impaired areas appeared to be general mood, consciousness, and a combination of different behavioral and neurological symptoms. *IQSEC2* variants were found to be slightly more than half as de novo and about one-fourth as maternally inherited, with about one-tenth of undefined origin, since mother’s data were unavailable for Patient ID #3, whereas genetic analysis for the parents of Patient ID #18 was in progress. In the analyzed case series, patients harboring inherited *IQSEC2* variants did not differ from *IQSEC2* patients with de novo variants with the exception of earlier diagnosis age of *IQSEC2* patients harboring inherited variants. Indeed, *IQSEC2* inheritance seems to impact only on age at diagnosis, although not related to clinical hallmark features and development milestones. Furthermore, female *IQSEC2* patients with de novo variants show a significantly higher prevalence of mandatory co-sleeping behavior as compared with their inherited variant female counterpart.

Interestingly, about one-third of patients harbored variants in other genes. The role of these reported gene variants, theoretically resulting in a mixed severity landscape interpretation, is to be further investigated. Moreover, given that our data were based on genetic reports coming from different laboratories and from different diagnostic genetic testing, the number and the pathogenetic weight of the gene variants remain unclear to date, and are an unavoidable limitation of the present study. In two patients, more than two gene variants were reported. A large number of mutated genes (about *n* = 66) have been previously linked to RTT-like phenotype [44,45]. Interestingly, about 6% of these genes can be associated with febrile seizures [44]. Of note, variants in the *SMARCC1* gene are reportedly associated with the RTT-like phenotype, as well as congenital hydrocephalus [87]. SMARCC1 is a member of the SWI/SNF family of proteins, whose members display helicase and ATPase activities and which are thought to regulate transcription of certain genes by altering the chromatin structure. The double pathogenic variant in two RTT-like phenotype linked genes i.e., *IQSEC2* and *SMARCC1* could have led to a more severe somatic growth deficiency, early onset of stereotypic hand movements.

Sleep disorders such as early awakening and insomnia have been previously reported in about one-sixth of *IQSEC2* patients [12]. In the present study, by using a validated structured semi-quantitative questionnaire [58], a remarkably higher prevalence (68.4%, 77.8% in males and 60% in females) of altered sleep emerged. Since sleep is a more frequent finding than previously thought, this aspect should be further investigated given our observed impact on the quality of life of these patients/families. Moreover, this feature relates with prior findings suggesting that sleep problems are important to explore among children with mental health issues, since it has been demonstrated that untreated sleep disturbances may lead to symptoms such as emotional lability, irritability, low tolerance to frustration, behavioral disorders, and aggressiveness that affect the daytime functioning of these children and complicate their management [88,89,90]. It is interesting to note that administration of sleep-influencing drugs was reported in a small fraction (i.e., one tenth) of the *IQSEC2* population. This observation was taken into consideration by the multidisciplinary team taking care of the patients. Although co-sleeping has been previously reported for patients with ASD [88], to the best of our knowledge, mandatory co-sleeping has not been reported before for this genetic NDD. From our survey, this feature appears to be linked to female gender, in which it was reported in half of the female patients. In the present study, co-mandatory co-sleeping was defined as a compulsory behavior feature springing directly from the patients as an indispensable condition in order to start and maintain the sleep. In contrast, co-sleeping as a parent choice springing from anxiety or need of closer monitoring was not considered.

In the present survey, mandatory co-sleeping was linked to de novo *IQSEC2* variants and reduced illness severity in *IQSEC2* female patients. These patients were younger, leaner, taller, and had less impaired social interaction. Bruxism was a more prevalent feature in this sub-cohort of *IQSEC2* patients. Since this feature has not been reported so far, more investigation on this feature, as well as sleep architecture, should be explored in the future. Among the most widely reported factors associated with a greater prevalence of co-sleeping in infants are socioeconomic factors like lower family income, lower maternal age, lower maternal education, and nonwhite ethnicity [91,92]. Literature on co-sleeping and sleep problems in children with mental health issues is scarce [88]. In children with ASD, sleep patterns, sleep problems, and their correlates show that co-sleeping is common in children with ASD, and these children experience a number of sleep issues [93].

Of interest, a not-negligible fraction of *IQSEC2* females have been misdiagnosed or initially suspected as RTT. We do believe that this is not a diagnostic mistake but rather an element for reflection. If these patients have been addressed toward a RTT diagnosis by experts in the field, this cannot be a plain mistake. Therefore, we searched for overlapping features in the *IQSEC2* female patients initially misdiagnosed as RTT. In the present study, we reported several phenotypical features in *IQSEC2* patients as a function of an initial RTT diagnosis. Overall, this peculiar cohort rarely behaves as a distinct and isolated patients’ cluster (i.e., illness severity score distribution). More often, several phenotypical signs and symptoms are indistinguishable from *IQSEC2* not initially diagnosed as RTT or typical RTT. Our data confirm a partial overlapping to the RTT phenotype [11]. In the present study, overlaps with the RTT phenotype exist, especially for the severity of neuro/behavioral features, as well as early impact on language and co-existence of OSAs. On the other hand, in our findings these patients do remarkably differ from RTT mainly for the lack of regression period and notably lower impact on growth, gastrointestinal involvement, and psychiatric comorbidity such as ASD-like features. Of interest, a remarkable difference concerns the hand stereotype pattern, which is very rarely on the midline. Differences from *IQSEC2* females and typical RTT girls extend to another aspect of illness severity, i.e., a reduced ANS involvement. From an epileptological point of view, this study is not focused on epilepsy itself (i.e., phenomenology, age-dependent variations), EEG findings, or types of AED therapy. We believe that this relevant issue deserves future, more specialistic research. From the extra-neurological point of view, *IQSEC2* females appear to exhibit less severe constipation as compared with the RTT group, whereas early developmental milestone delay seems to be more frequent in the *IQSEC2* population.

In the present study, an earlier age at menarche is reported for the first time in the *IQSEC2*-mutated female population based on the data from the survey. Altered pubertal timing, both early and late puberty, is often recognized as a constituent phenotype in patients presenting with multisystem syndromic developmental disorders with varied genetic causes [94]. The role of genetics in central precocious puberty has been previously underlined in RTT, Russell–Silver syndrome, Prader–Willi syndrome, MeCP2 duplication syndrome, Temple syndrome, Neurofibromatosis type I, and Williams–Beuren syndrome [95].

The relationship between *IQSEC2*-related encephalopathy and precocious puberty is unclear to date. Apart from a single description as an adverse event linked to risperidone treatment [1], precocious puberty has been reported as anecdotal evidence in a single patient with a de novo *IQSEC2* truncating variant [10]. Whether this feature is a component of the *IQSEC2* phenotypical spectrum remains, however, uncertain. One more affected *IQSEC2* female with precocious puberty (age 8 years) has been described and likely related to pituitary microadenoma [21]. At least one more report of an affected *IQSEC2* girl with early-onset puberty exists [9]. These sporadic reports are in striking contrast with the reported figure of 30 to 50% patients with precocious puberty phenotype related to *IQSEC2* gene alterations indicated on a rare disease online UK link [96]. Although no cases of precocious puberty were reported in our Italian survey, our observation of an earlier age at menarche in Italian patients harboring different *IQSEC2* pathogenic variants without significant relation with coexisting concomitant medication with risperidone nor demonstrated pituitary microadenoma adds more value to prior anecdotal data [1,10,21] and suggests that puberty time regulation is clinical topic of interest to be further explored within the *IQSEC2* phenotypical spectrum. Interestingly, precocious puberty has previously been associated with RTT and MDS in males, and more than 25% of RTT girls are known to initiate puberty early, even if the median age at menarche falls in the physiological range [97].

Overall, from the interviews, a series of extra-neurological issues emerged that are certainly in need of being addressed. These findings reinforce the concept that proper medical assistance in a complex NDD such as *IQSEC2*-related encephalopathy, should include several professionals taking care of the several individual and family needs.

## 5. Conclusions

*IQSEC2*-related encephalopathy is an exceedingly rare, but likely underdiagnosed, X-linked neurodevelopmental encephalopathy carrying a relevant neurological and extra-neurological clinical burden with an impact on life quality comparable or higher than that of RTT. Although limitations of the present study mainly reside in the way that the information was obtained, i.e., by structured interviews of parents/caregivers filtered by an expert physician instead of clinical direct examination, the results highlight the potential of focused clinical interviews whenever they are able to concentrate a patients’ population carrying an ultra-rare disease (<1:50,000) [98]. From one side, the information gathered by the interviews is able to pave the way towards new directions for early genetic diagnosis and windows of opportunity for prevention of epilepsy and raise new hypotheses on the underlying molecular mechanisms, and overall, on how the genotype translates into a phenotype in the *IQSEC2*-related encephalopathy. Likewise, this study shows that a focused survey is able to estimate the impact of a rare genetic disease on the life quality of the patients and their families, suggesting new avenues for intervention (i.e., sleep, puberty regulation in the affected females), while social needs are a relevant topic to be addressed in the near future.

## Figures and Tables

**Figure 1 children-10-01442-f001:**
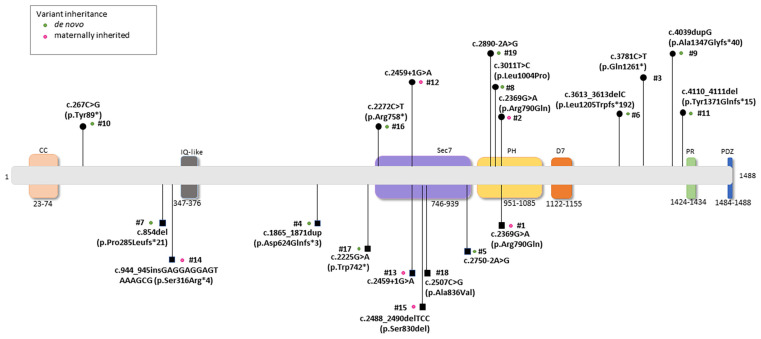
Schematic location and mutation severity of the pathogenic *IQSEC2* variants reported in the study. Legend for the corresponding protein domains: N-terminal coiled coil (CC), IQ calmodulin-binding motif (IQ), SEC7, Pleckstrin homology (PH), a potentially new 7th domain (D7), Proline-rich (PR), and PDZ-binding motif (PDZ). Male and female subjects harboring variants were indicated as squares and circles, respectively. Green dots: de novo occurrence; pink dots: maternal inheritance; no dots indicate that inheritance is unknown (i.e., mother’s data unavailable for Patient ID #3 and genetic analysis in progress for the parents of Patient ID #18).

**Figure 2 children-10-01442-f002:**
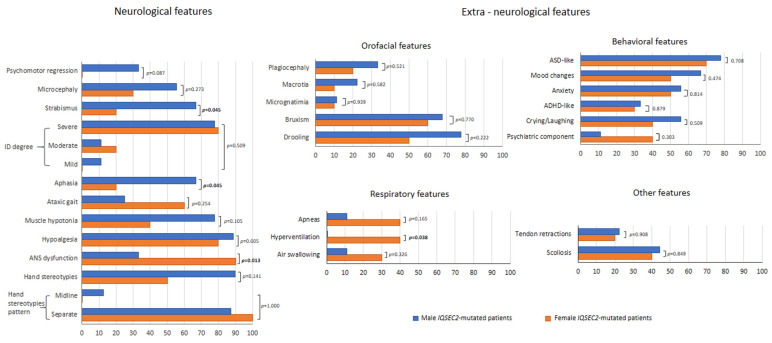
Clinical features (neurological and extra-neurological) of Italian *IQSEC2*-mutated patients as a function of gender.

**Figure 3 children-10-01442-f003:**
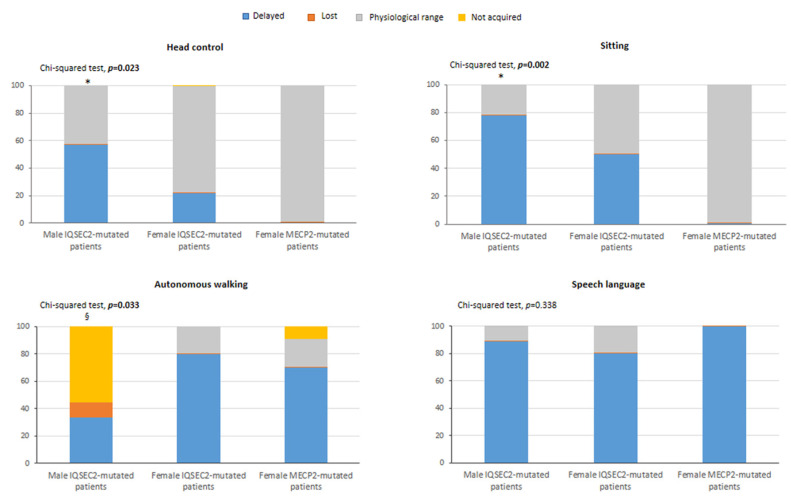
Key early development milestones in *IQSEC2*-mutated patients and typical RTT patients. Legend * *p* < 0.05 vs. RTT group. ^§^ comparison of “not acquired” component in male *IQSEC2*-mutated patients vs. RTT group *p* < 0.05. For male *IQSEC2*-mutated patients, head control and sitting: data not available for *n* = 2 patients. For female *IQSEC2*-mutated patients, head control: data not available for *n* = 1 patient.

**Figure 4 children-10-01442-f004:**
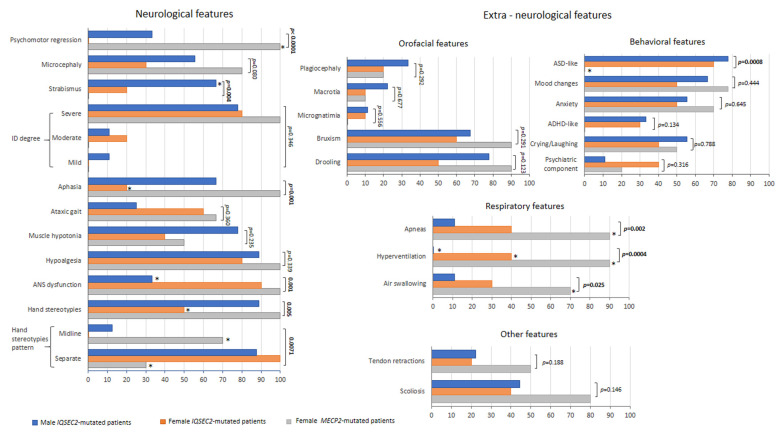
*IQSEC2* vs. RTT comparisons: clinical features (neurological and extra-neurological). Legend: * *p* < 0.05 vs. other groups.

**Figure 5 children-10-01442-f005:**
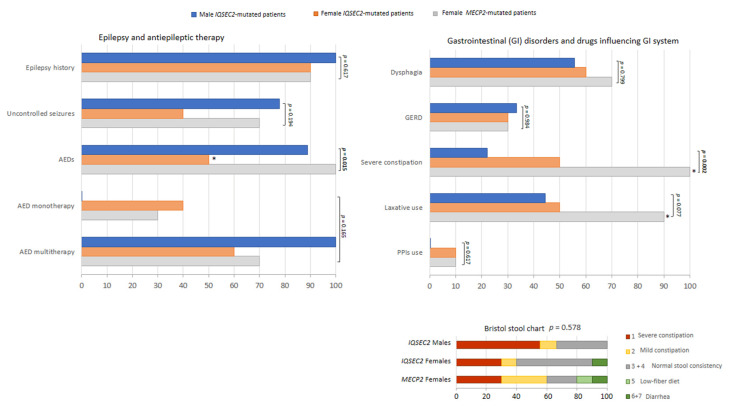
*IQSEC2* vs. RTT comparisons: epilepsy and GID. Legend: GERD, gastrointestinal reflux disease. * *p* < 0.05 vs. other groups.

**Figure 6 children-10-01442-f006:**
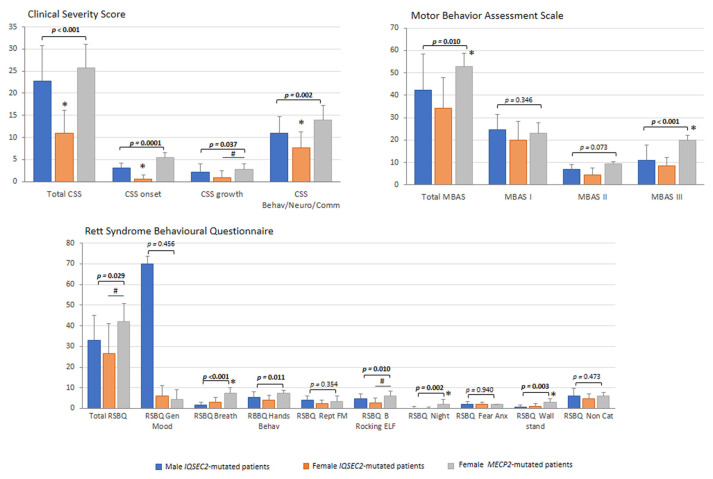
*IQSEC2* vs. RTT comparisons: illness severity. Legend: RSBQ Gen Mood, RSBQ general mood; RSBQ Breath, RSBQ breathing abnormalities; RSBQ Hands Behav, RSBQ hand behavior; RSBQ Rept FM, RSBQ repetitive face movements; RSBQ BRocking ELF, RSBQ body rocking and expressionless face; RSBQ Night, RSBQ night-time behavior; RSBQ FearAnx, RSBQ fear/anxiety; RSBQ Walkstand, RSBQ walking/standing; RSBQ NonCat, RSBQ items not categorized. * *p* < 0.05 vs. other groups. ^#^
*p* < 0.05 *IQSEC2* females vs. RTT patients.

**Figure 7 children-10-01442-f007:**
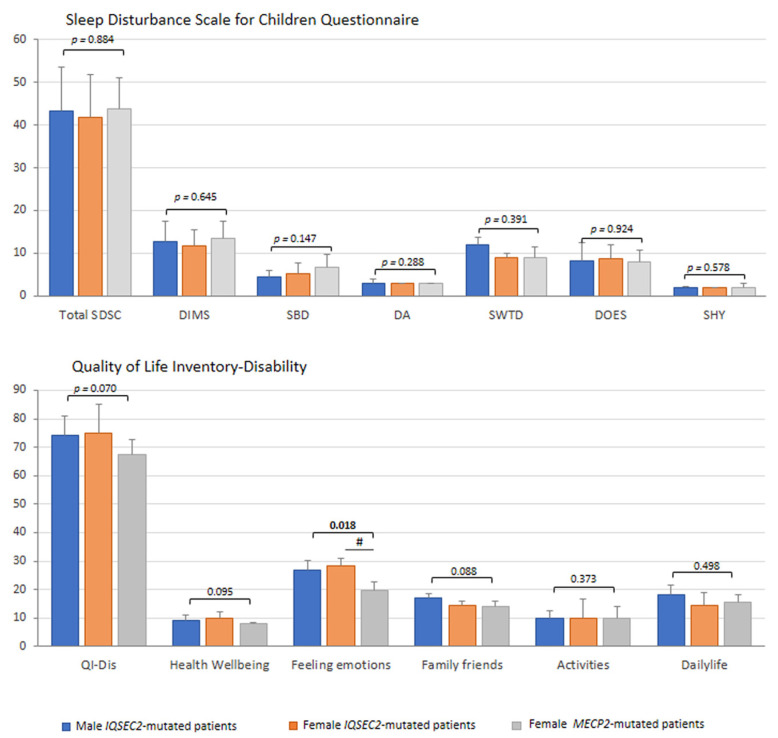
*IQSEC2* vs. RTT comparisons: sleep disturbance and life quality. Legend: SDSC, Sleep Disturbance Scale for Children Questionnaire; DIMS, difficulty in initiating and maintaining sleep factor; SBD, sleep breathing disorders; DA, disorders of arousals/nightmares; SWTD, sleep/wake transition disorders; DOES, disorders of excessive somnolence; SHY, sleep hyperhidrosis; QI-Dis, Quality of Life Inventory-Disability. ^#^
*p* < 0.05 *IQSEC2* females vs. RTT patients.

**Figure 8 children-10-01442-f008:**
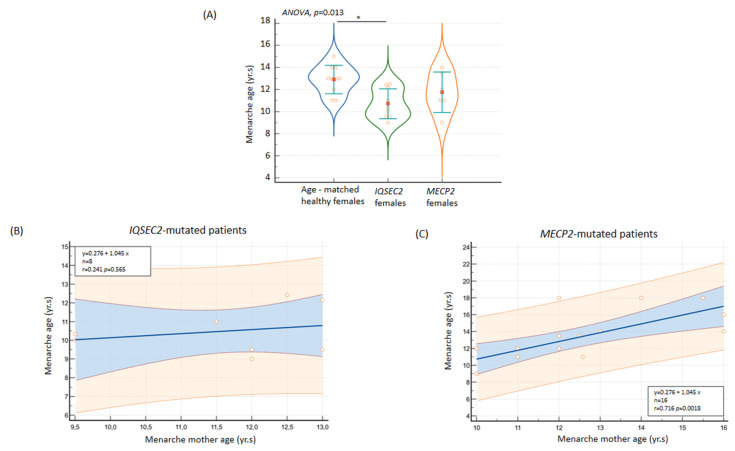
(**A**) Early menarche age in IQSEC2-mutated female patients as compared to healthy population and Rett syndrome. No relationship was observed between menarche age of patients and their mothers in IQSEC2-related encephalopathy (**B**) as compared to a demonstrated positive association in the RTT population (**C**). * *p* < 0.05.

**Figure 9 children-10-01442-f009:**
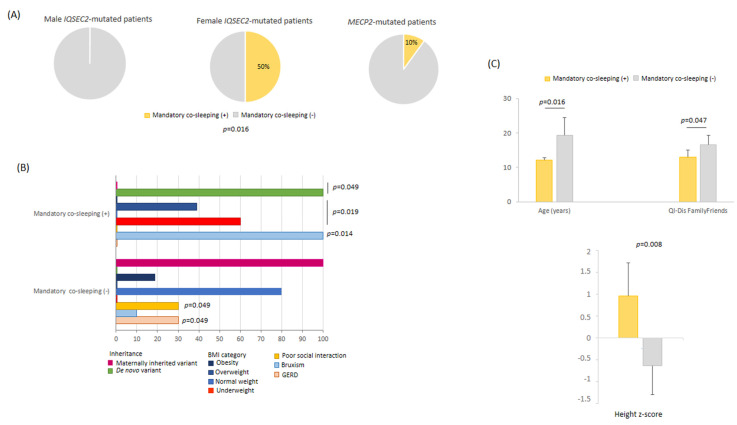
(**A**) Mandatory co-sleeping as a key behavioral feature in *IQSEC2*-mutated female patients. (**B**) Distinctive features as a function of mandatory co-sleeping in female *IQSEC2*-mutated patients. (**C**) *IQSEC2*-mutated females with mandatory co-sleeping are younger, taller, and have good social interaction as compared to those not showing mandatory co-sleeping.

**Figure 10 children-10-01442-f010:**
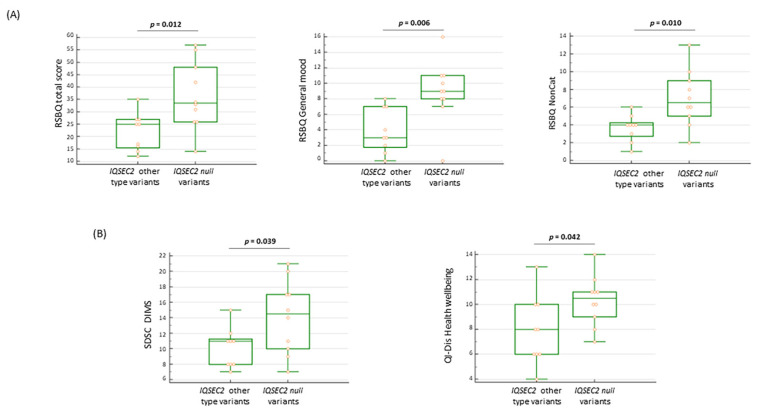
(**A**) Patients with *IQSEC2* null variants present more severe behavioral features (**B**) Disorders in initiating and maintaining sleep (SDSC DIMS) and altered perception of health wellbeing (QI-Dis Health wellbeing) in *IQSEC2*-patients with null variants. Legend: RSBQ NonCat = RSBQ items not categorized; SDSC DIMS = disorders in initiating and maintaining sleep; and QI-Dis Health wellbeing = Quality of Life Inventory-Disability health wellbeing.

**Figure 11 children-10-01442-f011:**
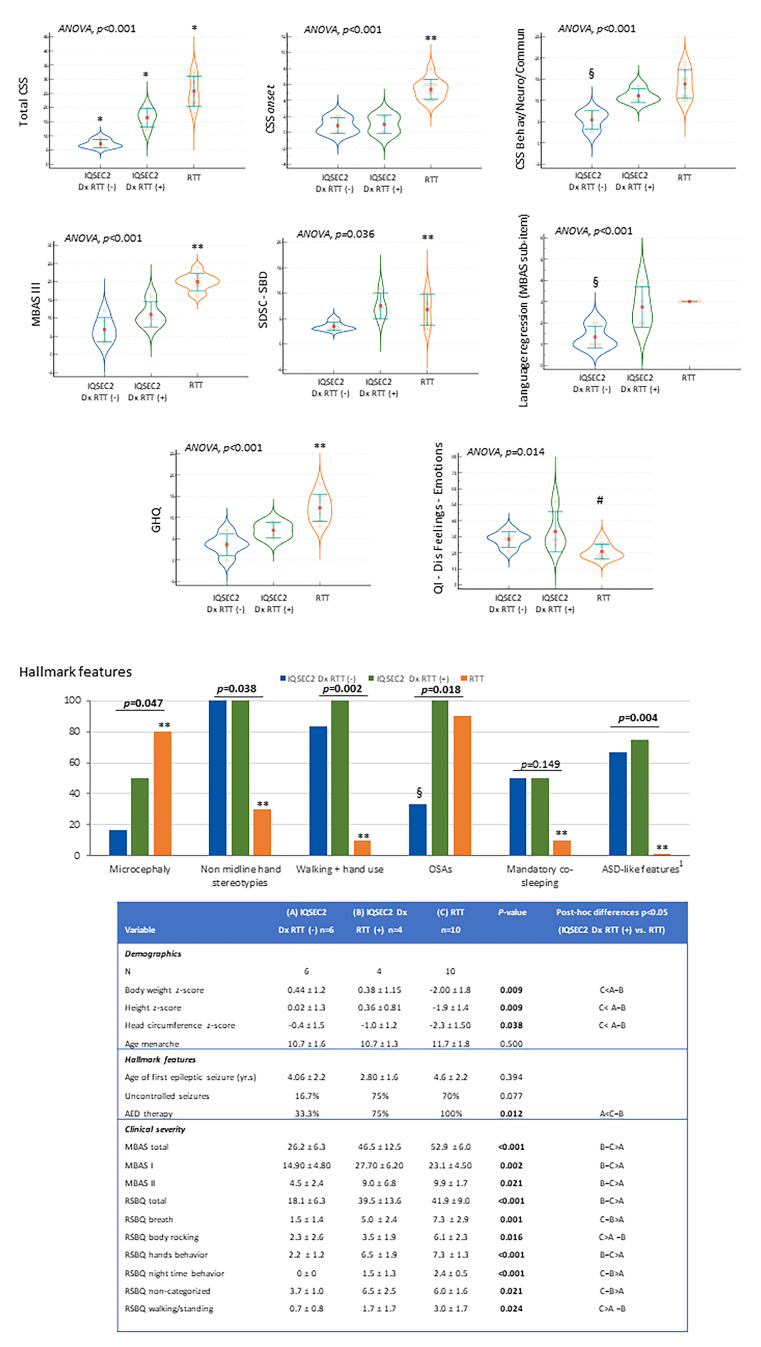
Comparisons between *IQSEC2* patients with initially misdiagnosed as RTT, *IQSEC2* patients not initially diagnosed as RTT, and typical RTT patients: key distinctive elements. Legend: Dx RTT (−), not initially diagnosed as RTT; Dx RTT (+), initially diagnosed as RTT; CSS, Clinical Severity Score; MBAS III, Motor Behavior Assessment Scale III (motor abilities/physical signs); SDSC-SBD, sleep breathing disorders subdomain of SDSC questionnaire; GHQ, gastrointestinal heath questionnaire; QI-Dis Feeling-Emotions, Feeling-Emotions subdomain of QI-Disability; AED, antiepileptic; ASD, autism spectrum disorders; GID, gastrointestinal disorders; OSAs, obstructive sleep apnea syndrome; RSBQ, Rett Syndrome Behavior Scale. * *p* < 0.05 vs. both other two groups, ** *p* < 0.05 RTT girls vs. *IQSEC2* Dx RTT (+), RTT girls vs. *IQSEC2* Dx RTT (−). ^§^
*p* < 0.05 *IQSEC2* Dx RTT (−) vs. RTT girls, *IQSEC2* Dx RTT (−) vs. *IQSEC2* Dx RTT (+). ^#^
*p* < 0.05 RTT girls vs. *IQSEC2* Dx RTT (+). ^1^ before the statically significance *p*-value. 1 only permanent ASD-like symptoms.

**Table 1 children-10-01442-t001:** Demographic and clinical data for Italian *IQSEC2*-mutated patients.

Variable	*IQSEC2*-Mutated Whole Population (19)	*IQSEC2*-Mutated Males (9)	*IQSEC2*-Mutated Females (10)	Comparison between *IQSEC2* M vs. F *p*-Value
**Demographics**				
N	19	9	10	
Age (years)	13.7 ± 7.3 (range 2.3–34)	12.7 [3.3–15.8]	13.1 [12.1–19.4]	0.094
Age category				
Pediatric	9 (47.4%)	5 (55.6%)	4 (40%)	0.588
Adolescent	6 (31.6%)	3 (33.3%)	3 (30%)	
Adult	4 (21.1%)	1 (1.11%)	3 (30%)	
Genetic diagnosis ≤ 2 years	1/19 (5.3%)	1/9 (11.1%)	0 (0%)	0.292
Body weight z-score	−0.18 ± 1.37	−0.86 ± 1.37	0.42 ± 1.11	0.040
Height z-score	−0.15 ± 1.14	−0.49 ± 1.17	−0.16 ± 1.08	0.223
Head circumference z-score	−0.92 ± 1.03	−1.22 ± 0.79	−0.65 ± 1.17	0.232
BMI z-score	−0.29 ± 1.47	−0.87 ± 1.37	0.24 ± 1.42	0.101
**Perinatal/neonatal data**				
Gestational age (weeks)	38.5 ± 2.7	40 [38.0–40.2]	38 [37.0–40.0]	0.318
Prematurity	3 (15.8%)	1 (11.1%)	2 (20%)	1.000
Adverse perinatal events	6 (31.6%)	2 (22.2%)	4 (40%)	0.418
Neonatal birth weight (g)	3120.0 ± 513.4	3180.0 ± 605.2	3066.8 ± 441.0	0.452
Neonatal birth weight z-score	0.03 ± 1.11	−0.21 ± 0.93	0.26 ± 1.25	0.367
Birth weight category				
AGA	14 (73.7%)	7 (77.8%)	7 (70%)	0.869
LGA	3 (15.8%)	1 (11.1%)	2 (20%)	
SGA	2 (10.5%)	1 (11.1%)	1 (10%)	
Delivery mode				
SVD	13 (68.4%)	6 (66.7%)	7 (70%)	0.219
ELCS	2 (10.5%)	2 (22.2%)	0 (0%)	
EMCS	4 (21.1%)	1 (11.1%)	3 (30%)	
Breastfeeding	11 (57.9%)	4 (44.4%)	7 (70%)	0.273
Breastfeeding duration (months)	10.8 ± 10.6	8.5 [6.0–16.5]	12 [6.1–12.0]	0.719
**Key early developmental milestones**				
*Head control*				
Delayed	6 (37.5%)	4 (57.1%)	2 (22.2%)	0.166
Physiological range	10 (62.5%)	3 (42.9%)	7 (77.8%)	
*Autonomous sitting*				
Delayed	12 (63.2%)	7 (77.8%)	5 (50%)	0.222
Physiological range	7 (36.8%)	2 (22.2%)	5 (50%)	
*Autonomous walking*				
Delayed	11 (57.9%)	3 (33.3%)	8 (80%)	0.017
Lost	1 (5.3%)	1 (11.1%)	0 (0%)	
Physiological range	2 (10.5%)	0 (0%)	2 (20%)	
Not acquired	5 (26.3%)	5 (55.6%)	0 (0%)	
*Speech*				
Delayed or regressed	16 (84.2%)	8 (88.9%)	8 (80%)	0.606
Physiological range	3 (15.8%)	1 (11.1%)	2 (20%)	

Data are expressed as mean ± SD or median [inter-quartile range IQR], as appropriate. Legend: BMI, body mass index; AGA, appropriate for gestational age; LGA, large for gestational age; SGA, small for gestational age; SVD, spontaneous vaginal delivery; ELCS, elective caesarean section; EMCS, emergency caesarean section.

**Table 2 children-10-01442-t002:** IQSEC2 vs. RTT comparisons: demographic data.

Variable	*A* *IQSEC2*-Mutated Males	*B* *IQSEC2*-Mutated Females	*C* *MECP2*-Mutated Females	Comparison between Groups (*p*-Value)	Post Hoc Differences *p* < 0.05
**Demographics**					
N	9	9	10		
Age (years)	10.7 ± 6.6	16.4 ± 7.2	15.5 ± 7.7	0.210	
Age category					
Pediatric	9 (47.4%)	5 (%)	4 (%)	0.588	
Adolescent	6 (31.6%)	3 (%)	3 (%)		
Adult	4 (21.1%)	1 (%)	3 (%)		
Genetic diagnosis ≤ 2 years					
	2 (22.2%)	0 (0%)	8 (80%)	0.0005	C > A ~ B
Body weight z-score	−0.86 ± 1.37	0.42 ± 1.11	−2.00 ± 1.76	0.004	C < B ~ A
Height z-score	−0.49 ± 1.17	−0.16 ± 1.08	−1.90 ± 1.44	0.003	C < B ~ A
Head circumference z-score	−1.23 ± 0.79	−0.65 ± 1.17	−2.3 ± 1.5	0.015	C < B ~ A
BMI z-score	−0.87 ± 1.37	0.24 ± 1.42	−1.26 ± 1.41	0.062	
**Perinatal/neonatal data**					
Gestational age (weeks)	40 [38.0–40.2]	38 [37.0–40.0]	40 [40.0–40.0]	0.250	
Prematurity	1 (11.1%)	2 (20%)	0 (0%)	0.339	
Adverse perinatal events	2 (22.2%)	4 (40%)	1 (10%)	0.289	
Neonatal birth weight (g)	3180.0 ± 605.2	3066.8 ± 441.0	3187.0 ± 223.4	0.795	
Neonatal birth weight z-score	−0.21 ± 0.93	0.26 ± 1.25	−0.34 ± 0.36	0.379	
Birth weight classification					
AGA	7 (77.8%)	7 (70%)	9 (90%)	0.698	
LGA	1 (11.1%)	2 (20%)	0 (0%)		
SGA	1 (11.1%)	1 (10%)	1 (10%)		
Delivery mode					
SVD	6 (66.7%)	7 (70%)	8 (80%)	0.249	
ELCS	2 (22.2%)	0 (0%)	0 (0%)		
EMCS	1 (11.1%)	3 (30%)	2 (20%)		
Breastfeeding	4 (44.4%)	7 (70%)	10 (100%)	0.025	C < B ~ A
Breastfeeding duration (months)	9.7 ± 5.2	11.3 ± 6.2	10.8 ± 7.2	0.927	

Data are expressed as mean ± SD or median [inter-quartile range IQR] as appropriate. Legend: BMI, body mass index; AGA, appropriate for gestational age; LGA, large for gestational age; SGA, small for gestational age; SVD, spontaneous vaginal delivery; ELCS, elective caesarean section; EMCS, emergency caesarean section. Data not available for *n* = 2 patients. Data not available for *n* = 1 patient.

**Table 3 children-10-01442-t003:** Comparisons between *IQSEC2*-mutated patients and typical *MECP2*-mutated RTT: diagnostic criteria for RTT [30] as applied to the examined *IQSEC2*-patients.

Variable	Italian *IQSEC2* Population (ID Patient #)																		
	1	2	3	4	5	6	7	8	9	10	11	12	13	14	15	16	17	18	19
Gender	M	F	F	M	M	F	M	F	F	F	F	F	M	M	M	F	M	M	F
Regression followed by recovery or stabilization	−	−	−	+	+	−	+	−	+	−	−	−	−	−	−	−	−	−	−
*Main criteria*																			
Partial/complete loss of acquired purposeful hand skills	+	+	−	+	+	+	+	+	−	−	+	−	+	+	+	+	+	−	−
Partial or complete loss of acquired spoken language	+	+	−	+	+	+	+	+	+	+	+	+	+	+	−	+	+	+	+
Gait abnormalities	+	−	−	+	+	+	+	+	+	+	−	+	+	+	−	−	+	+	+
Stereotypic hand movements	+	−	+	+	+	+	+	+	+	+	+	−	+	+	−	+	+	+	−
Total criteria (*n*)	4	2	1	4	4	4	4	3	3	3	3	2	4	4	1	3	4	3	2
*Exclusion criteria*																			
Secondary brain injury	−	−	−	−	−	−	−	−	−	−	−	−	−	−	−	−	−	−	−
Grossly abnormal PM development in first 6 months	−	−	−	+	+	−	+	−	+	+	+	−	+	+	+	+	+	+	−
Total exclusion criteria (*n*)	−	−	−	−	+	−	+	+	+	+	+	−	+	+	+	+	+	+	−
*Supportive criteria*																			
Breathing disturbances when awake	−	−	+	−	−	−	−	+	+	−	+	−	−	−	−	−	−	−	−
Bruxism when awake	+	−	−	+	−	+	+	−	−	+	+	+	−	−	+	+	+	+	+
Impaired sleep pattern	+	−	+	+	−	+	+	+	+	−	−	−	+	+	−	+	+	+	+
Abnormal muscle tone	+	−	−	+	+	+	−	+	−	−	−	+	+	+	−	+	+	+	−
Peripheral vasomotor disturbances	−	−	+	+	−	+	+	+	+	+	+	+	+	−	−	+	−	−	+
Scoliosis/kyphosis	+	−	+	+	−	−	−	+	−	−	−	+	+	−	−	+	+	−	−
Growth retardation	+	−	−	+	+	−	−	−	−	−	−	−	−	−	−	−	−	−	−
Small cold hands and feet	−	−	−	−	−	−	−	−	−	−	−	−	−	−	−	−	−	−	−
Inappropriate laughing/screaming spells	−	−	+	+	−	+	+	−	+	−	−	−	−	+	+	+	−	+	+
Diminished response to pain	+	−	+	+	+	+	+	+	+	−	+	+	+	+	+	+	+	−	−
Eye pointing	−	−	+	−	+	+	+	−	−	+	+	−	−	+	+	−	+	+	+
Total supportive criteria (*n*)	6	−	7	8	4	7	6	6	5	3	5	5	5	5	4	7	6	5	4

## Data Availability

The raw data supporting the conclusions of this article will be made available by the authors, without undue reservation.

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
