# Peer review of "Natural Course of IQSEC2-Related Encephalopathy: An Italian National Structured Survey"

_children, 2023, doi:10.3390/children10091442_

Round 1

Reviewer 1 Report

The authors present a thorough phenotypic analysis of a rare genetic cause of intellectual disability, IQSEC2-related encephalopathy. This careful and well-researched description of 19 patients will help to further understanding of this underdiagnosed disorder.

Note: I did not find the supplementary figures included in either the main manuscript pdf or the "non-published material" pdf.

A few specific comments:

Section 3.1 - consider displaying all incidence and prevalence figures in the same format (e.g. 0.07 per 100,000 for IQSEC2, 3.2 per 100,000 for US RTT) to allow for direct comparisons.

Section 3.2 - as noted above, supplemental table S2 was not available for review. The term "co-mutation" is imprecise as every individual carries numerous mutations in most of our genes; what you are referring to is a pathogenic variant in an additional gene. Later, in the discussion (lines 775-776) the authors state, "Interestingly, about one third of patients harbored coexisting co-mutations in other genes." However, this rate will vary greatly depending on the diagnostic testing utilized (e.g. gene panel vs exome vs genome sequencing).

Figure 1 - please explain how the p.Trp742* nonsense variant is considered a mild mutation; one would expect it to be considered severe as are the other nonsense and frameshift variants.

There are a few awkward sentences that need rephrasing (e.g. lines 67-68).

Reviewer 2 Report

This is a manuscript about an Italian National structured survey about natural course of the IQSEC2-related encephalopathy, in which most information was obtained through the interview of parents/caregivers. Although this study contains too vast amount of data, it seems informative and educational for diagnosing IQSEC2-related encephalopathy, sharing the clinical features with RTT syndrome.

There are a few comments;

1. L.273 and 580; What is the definition for IQSEC mutation severity classification. Although the author refers two papers, #47 and 48, it is difficult to find it. Is the severe mutation a variant that cause severe phenotype, or a variant of nonsense or splicing?

2. L.627; Is “RTT and RTT patients” incorrectly spelled as RTT and RTT-like patients?
